# Inverse Estuaries in West Africa: Evidence of the Rainfall Recovery?

Luc Descroix [1,2,*], Yancouba Sané [2,3], Mamadou Thior [2,3], Sylvie-Paméla Manga [2,3,4], Boubacar Demba Ba [2,3], Joseph Mingou [2,3], Victor Mendy [3,5], Saloum Coly [3,5], Arame Dièye [2,3], Alexandre Badiane [1,2,3], Marie-Jeanne Senghor [2], Ange-Bouramanding Diedhiou [2], Djiby Sow [2,3], Yasmin Bouaita [2], Safietou Soumaré [2,6,7], Awa Diop [2,8], Bakary Faty [9], Bamol Ali Sow [3,5], Eric Machu [5,10], Jean-Pierre Montoroi [11], Julien Andrieu [2,7] and Jean-Pierre Vandervaere [12]

[1] IRD UMR PALOC MNHN/IRD/Sorbonne-Université, 75231 Paris, France; a.badiane785@zig.univ.sn
[2] LMI PATEO, UGB, BP 234 Saint Louis, Senegal; saneyancouba@gmail.com (Y.S.); thioryaz@yahoo.fr (M.T.); s.manga4555@zig.univ.sn (S.-P.M.); bademba ba@gmail.com (B.D.B.); josephmingou30@gmail.com (J.M.); dieyearame91@gmail.com (A.D.); mjsenghor@gmail.com (M.-J.S.); bouramanding@gmail.com (A.-B.D.); sowsowdjiby@gmail.com (D.S.); yasmin.bouaita@gmail.com (Y.B.); safisoumare1@gmail.com (S.S.); a.diop754@zig.univ.sn (A.D.); Julien.ANDRIEU@univ-cotedazur.fr (J.A.)
[3] UASZ Université Assane Seck de Ziguinchor, BP 523 Ziguinchor, Senegal; victormendy97@yahoo.fr (V.M.); colysaloum@gmail.com (S.C.); bsow@univ-zig.sn (B.A.S.)
[4] Université de Lorraine, UFR des Sciences Humaines et Sociales, 54015 Nancy, France
[5] LMI ECLAIRS, UASZ, ENS, BP 5036 Dakar, Senegal; eric.machu@ird.fr
[6] LaSTEE, Ecole Polytechnique de Thiès, DPA 10 Thiès, Senegal
[7] ESPACE Lab, Université Côte d'Azur, UFR Espaces & Cultures Campus, 06204 Nice, France
[8] Université Versailles St Quentin en Yvelines, UFR Sciences Sociales, 78280 Guyancourt, France
[9] DGPRE, Direction de la Gestion et la Planification des Ressources en Eau, section 2, 20000 Diamniadio, Senegal; bakaryfaty@gmail.com
[10] IRD/Laboratoire d'Océanographie Physique et Spatiale (LOPS), IUEM, Univ. Brest, CNRS, IRD, Ifremer, 29280 Plouzané, France
[11] IRD/IEES; Institut d'écologie et des sciences de l'environnement, 93143 Bondy, France; jean-pierre.montoroi@ird.fr
[12] IGE/Université Grenoble Alpes; Institut des Géosciences et Environnement, 38058 Grenoble, France; jean-pierre.vandervaere@univ-grenoble-alpes.fr
* Correspondence: luc.descroix@ird.fr

**Abstract:** In West Africa, as in many other estuaries, enormous volumes of marine water are entering the continent. Fresh water discharge is very low, and it is commonly strongly linked to rainfall level. Some of these estuaries are inverse estuaries. During the Great Sahelian Drought (1968–1993), their hyperhaline feature was exacerbated. This paper aims to describe the evolution of the two main West African inverse estuaries, those of the Saloum River and the Casamance River, since the end of the drought. Water salinity measurements were carried out over three to five years according to the sites in order to document this evolution and to compare data with the historical ones collected during the long dry period at the end of 20th century. The results show that in both estuaries, the mean water salinity values have markedly decreased since the end of the drought. However, the Saloum estuary remains a totally inverse estuary, while for the Casamance River, the estuarine turbidity maximum (ETM) is the location of the salinity maximum, and it moves according to the seasons from a location 1–10 km downwards from the upstream estuary entry, during the dry season, to a location 40–70 km downwards from this point, during the rainy season. These observations fit with the functioning of the mangrove, the West African mangrove being among the few in the world that are markedly increasing since the beginning of the 1990s and the end of the dry period, as mangrove growth is favored by the relative salinity reduction. Finally, one of the inverse estuary behavior factors is the low fresh water incoming from the continent. The small area of the Casamance and Saloum basins

(20,150 and 26,500 km$^2$ respectively) is to be compared with the basins of their two main neighbor basins, the Gambia River and the Senegal River, which provide significant fresh water discharge to their estuary.

**Keywords:** water salinity; inverse estuaries; West Africa; drought; mangrove

## 1. Problematics, State of the Art

West African Sahelian and Sudanian areas commonly have very flat coast plains. Therefore, rivers have long estuaries in front of their low hydraulicity. The Gambia River estuary, 450 km long, is considered the second longest in the world after that of the Amazon River, although its basin has an area 100 times and a discharge 1000 times smaller. In West Africa, only great basins provide enough fresh water to reach pushing saline waters seasonally out of the river mouths. In this constraining context, it is very easy for marine salt water to enter profoundly in the continent, and it is also easy for the tide to influence deeply the river water level within the continent. An inverse estuary is an estuary in which freshwater input is less than the losses due to evaporation; such estuaries contain hyperhaline water (e.g., Laguna Madra in Texas; [1]). Inverse estuaries were also defined by Pritchard [2] as the ones where salinity increased with distance to the mouth. In an inverse estuary, sea water enters the estuary from downstream to upstream to compensate for the losses due to evaporation, carrying salt, which raises concentration [3].

Some of these estuaries have very low fresh water income and are located in areas with very high evaporation. This leads to an increase in water density and a hypersaline density downwelling. Therefore, a surficial water flow is noticed upwards, and another flow is noticed near the bottom downwards [4]. In the case of the Saloum River, the increase is observed to be in a roughly linear fashion with distance to the sea [5].

These inverse estuaries are commonly found in semiarid or arid areas, such as southern Australia [6], northern Australia [7], some areas of Texas [1], Baja California (NW Mexico) [8,9], and in the Sahel [10], among others. Such hypersaline estuaries are relatively rare on the Earth. Other examples are the Red Sea and the Persian Gulf in the northern Indian Ocean [11], Shark Bay and Exmouth Gulf in Western Australia [12], and the hypersaline Coorong estuary/lagoon in South Australia [13,14]. The characteristics of Spencer Gulf, South Australia, are that evaporation exceeds precipitation all year round and that the spring–neap tidal cycle is greatly exaggerated [6].

In West Africa, as in the other estuaries, enormous volumes of marine water are entering the continent [15]. Fresh water discharge is commonly strongly linked to rainfall level [5]. During the Great Sahelian Drought (1968–1993), some main rivers completely dried up; this was the case of the Niger River in Niamey in May 1985 [16]. In extreme cases, "a negative water budget has even more drastic effects on smaller rivers: discharge becomes negative, (and) seawater may invade the estuary which becomes hyperhaline" [17]. As an example, "the Casamance River estuary, in a dry year, and during the dry season, can be changed into an evaporation basin, concentrating marine salts coming from [the] Atlantic Ocean and becoming a threat to fluvio-marine areas soils" [17].

Southward Dakar Senegambian estuaries are subject to [this] "unusual hydrodynamical regime caused by weak or absent run-off" [18]. Such a process has been occurring in two coastal "rivers" of Senegal, the Casamance and the Saloum Rivers, which are both actually tide-influenced "inverse estuaries" [5]. Therefore, the West African Sudano-Sahelian coast is one of the regions in the world where inverse estuaries are observed [19].

West Africa has shown a great interdecadal rainfall variability. Few data before 1950 allow highlighting two dry periods in 1910–1915 and 1940–1944. After the Second World War, the number of rain gauges increased significantly, and the following evolution can be described [20–22]:

- a very rainy period from 1950 to 1967
- a long and very dry period from 1968 to 1993 (whole West Africa) and from 1968 to 1998 in Senegambia and Mauritania
- a rainfall recovered period (1994 to 2018 in WA, 1999 to 2018 in Senegambia), which was characterized by a rainfall annual amount close to the 1918–2017 average and an intensification of rainfall: an increase in rainfall intensity [23] and in the number of extreme rainfall events [24] are observed.

The evolution in coastal Senegal is similar to the regional one. Figure 1 shows the location of Senegambia (Figure 1a) and the location of rain gauge stations (Figure 1b). Figure 2 gives the rainfall evolution at different spatiotemporal scales. Figure 2a shows the evolution at three Senegalese coastal stations from 1917 to 2017, and Figure 2b shows the one in the inverse estuaries areas, southward from Dakar, from 1950 to 2017. Figure 2c,d give the SPI (Standard Precipitation Index) respectively for Senegambia and for the Casamance River basin. This confirms also the conclusions of Faye and Sané (2015; [25]), who observed the end of the long dry period in 1996 for the Casamance River Basin.

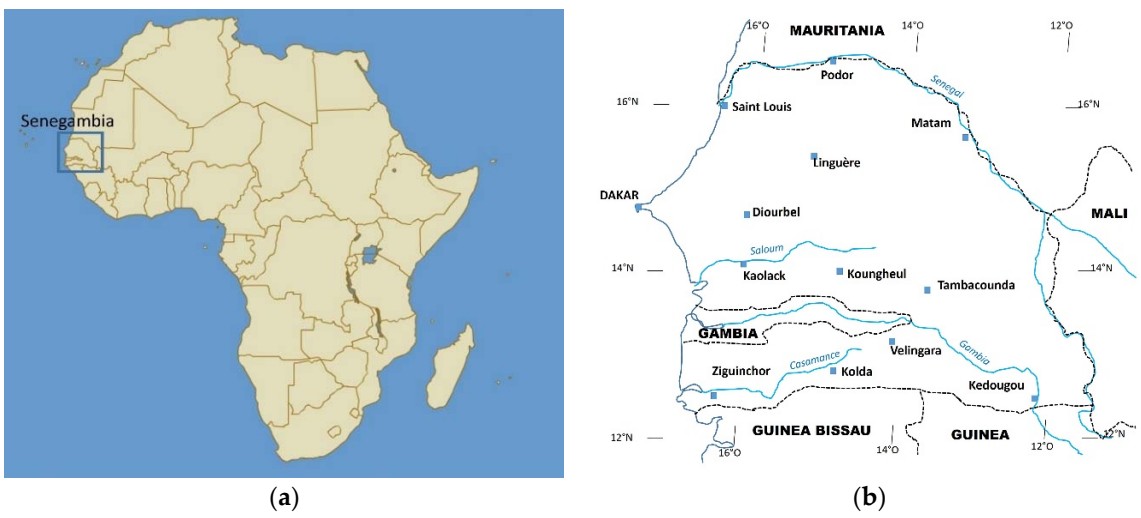

**Figure 1.** Location of Senegambia (**a**) and location of main rivers, boundaries, and rain gauge stations (**b**).

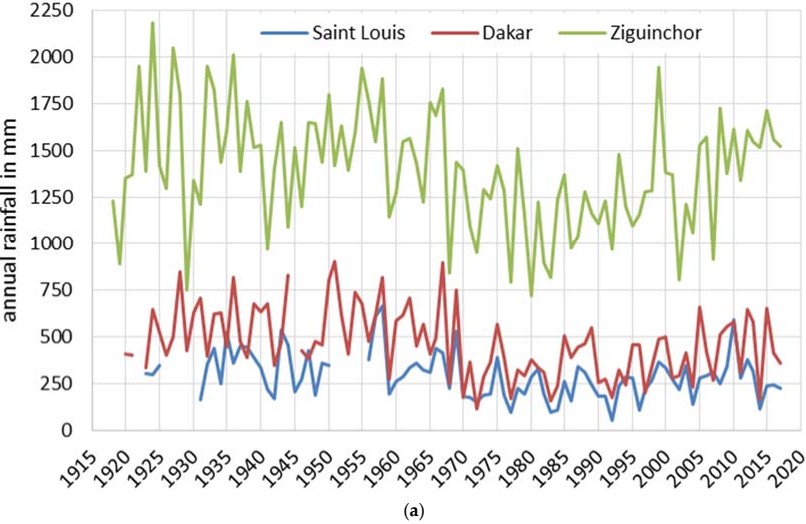

**Figure 2.** *Cont.*

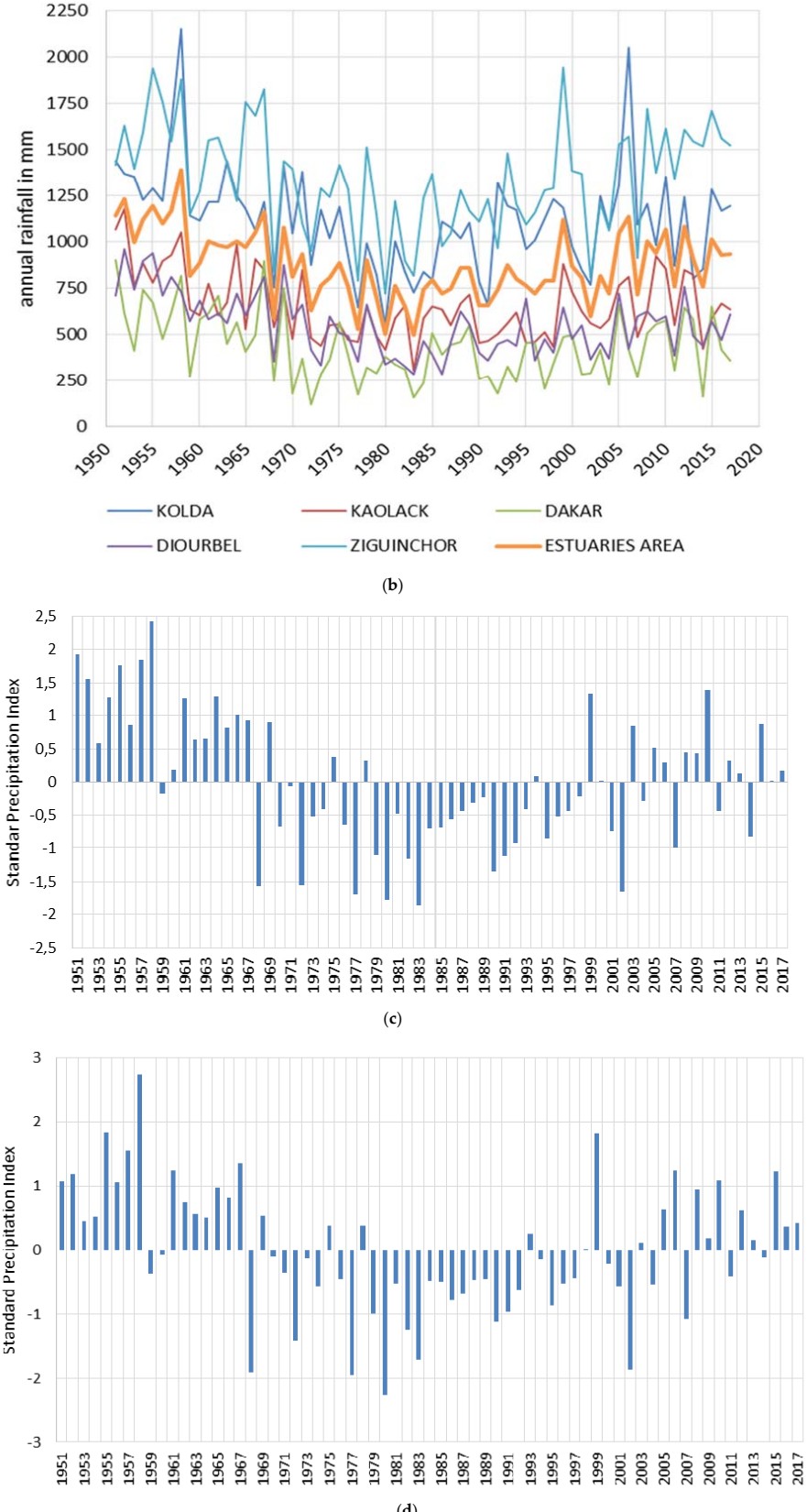

**Figure 2.** (**a**) Evolution of annual rainfall from 1917 to 2017 in three coastal Senegalese stations; (**b**) evolution of annual rainfall in the inverse estuary area; Rainfall Index Evolution in Senegambia (SPI, Standard Precipitation Index) (**c**), and in Casamance River basin: Ziguinchor, Kolda, Velingara (SPI Standard Precipitation Index) (**d**).

Teleconnections with the ITCZ (Inter Tropical Convergence Zone) fluctuations and their impact on flooding were analyzed by Mahé et al. (2013) [26].

When a river bed is close to the sea level, sea water may ingress during the low-water seasons. If the freshwater inflow is less than the loss through evaporation, salinity becomes higher than that in the sea. During the drought (1968–1998) Binet et al. 1995 [27] wrote, "This happens in the Senegal and the Casamance, particularly during the "current" drought".

Before 1960, marine water intrusions into the Senegal River during the dry season were negligible; but, in 1969 they reached 150 km upstream, and in 1978, they reached even 200 km [28]. Before the construction of the Diama (1986) and Manantali (1988) dams, high salinity values were observed especially during the dry period (1968–1986), particularly in the later years when salinity was concentrated due to a succession of very dry years in the second paxorysm of the Great Drought (1982–1986). Values above 35‰ were almost each year noticed at the end of the dry season (June and July) 80 km upstream from the mouth by Saos et al. (1984) [29]. These authors evidenced the inverse functioning of the estuary. Such behavior is widespread in the other Senegambian estuaries (Sine-Saloum, Gambia, and Casamance Rivers); however, it was strictly exceptional and of very short duration in the Senegal estuary [28]. These authors observed that the recovery of normal functioning is quite long (27 days in 1981 and 38 days in 1982).

Similar to the Casamance and Saloum estuaries, the semidiurnal microtidal Somone River estuary (80 km southeast of Dakar), where the maximum tidal range is about two meters, is characterized by an inverse salinity gradient [30,31]. Fluvial flows in the Somone estuary are null. Therefore, the salinity gradient is inverse; the only fresh water incomes observed during the 2007–2010 period are provided by rainfall and groundwater [31].

Hydrochemical analysis confirms that the Casamance River and Saloum River are "inverse estuaries", in which the water is salted during most of the year (at least 9 months) and hypersaline at the end of dry season [32].

Some estuaries within dry tropical areas (e.g., in Australia) show a combination of estuarian "normal" and "inverse" modes [4]. Inverse functioning of the Saloum River estuary, where salinity can reach 130 g/L [33], strongly affects fish and all marine species populations.

To conclude, Baran (1994) [34] summarized the context explaining that "Gambia river has a normal estuary, i.e., with decreasing salinity upwards. The Casamance estuary is inverse (rising salinity upwards) during the dry season and normal during the rainy season, while the Sine-Saloum is a "ria" where fluvial flows are null and the estuary is always inverse".

This study proposes to describe the current behavior of two West African inverse estuaries and to compare it with that observed during the Sahelian dry period of the end of the 20th century.

## 2. Methodology

In order to document the current functioning of the Saloum and the Casamance estuaries (see location in Figure 3), two measurement devices were implemented:

- a twice a year direct measuring campaign through the Saloum and the Casamance estuaries since the end of 2016:

This measuring campaign included a network of measurement sites in both Saloum and Casamance estuaries, salinity measures with a refractometer PCE© 0100 (67250 Soultz-sous-forêt, France) and, for values lower than 20 mS/cm, a conductimeter HANNA© HI 98130 (Woonsocket, RI, the USA), which also gives temperature and pH. This campaign was carried out at the end of the dry season (may) and at the end of the rainy season (November);

- a settled ensemble of five multi-sensor devices localized in the Casamance River estuary only, since Jan 2014;

These devices are CTD sensors (Conductivity, Temperature, Depth) model Decagon© CTD 10 (Pullman, WA, the USA), each one was coupled with a Decagon© EM50 data logger.

According to Noblet (2012) [35], salinity is calculated as follows:

$$Sa = (0.72 \times \sigma - 3.06) \times (1 + 0.02\,(T - 25)) \tag{1}$$

where:

- $S_a$ is salinity in psu,
- $\sigma$ is conductivity (mS/cm).
- T is temperature in °C.

Equation (2) was validated with the measured values with both a refractometer and field conductimeter. Since the bolons' water temperature always ranged between 21 and 28 °C, the deviation between measured and calculated values was low, rarely exceeding 2.5% (the highest observed difference was 4.8%).

Table 1 summarizes the data collected thanks to the implemented instruments.

**Table 1.** Data collected with Conductivity, Temperature, Depth (CTD) sensors around the Casamance estuary.

| Site * | Number of Measurements | Number of Averaged Points | % Missing Data | % Corrected Data |
|---|---|---|---|---|
| Karabane | 129,500 | 804 | 50 | 10 |
| Ziguinchor | 127,400 | 873 | 50 | 12 |
| Goudomp | 178,200 | 1653 | 20 | 10 |
| Baila | 214,700 | 2228 | 5 | |
| Niambalang | 168,000 | 1547 | 15 | 8 |

\* see location in Figure 3.

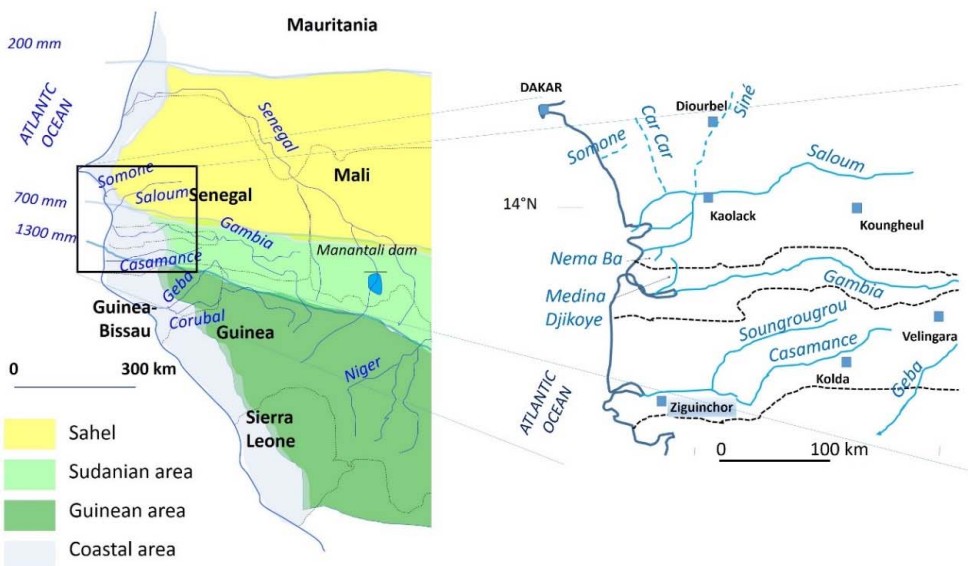

**Figure 3.** Location of cited West African estuaries. At **left**: geoclimatical areas (some annual rainfall amounts are indicated) and the main rivers network; at **right**: zoom on the study area at the local scale (dotted lines are the countries boundaries).

CTD data were regularly calibrated with that of the conductimeters (up to 20 mS/cm) and that of the refractometer (from 5 g/L, i.e., approximately 3 mS/cm, up to 100 mS/cm).

- Refractometers were calibrated with distillated water at the beginning and at the end of each measurement fraction of the day.

- Conductimeters were calibrated with standard dilution products supplied by the provider at the beginning and at the end of each measurement fraction of day in order to ensure the quality of measured data.

## 3. Results

Figure 4a (Saloum estuary) and 4b (Casamance estuary) give the location of the observations and implementation devices.

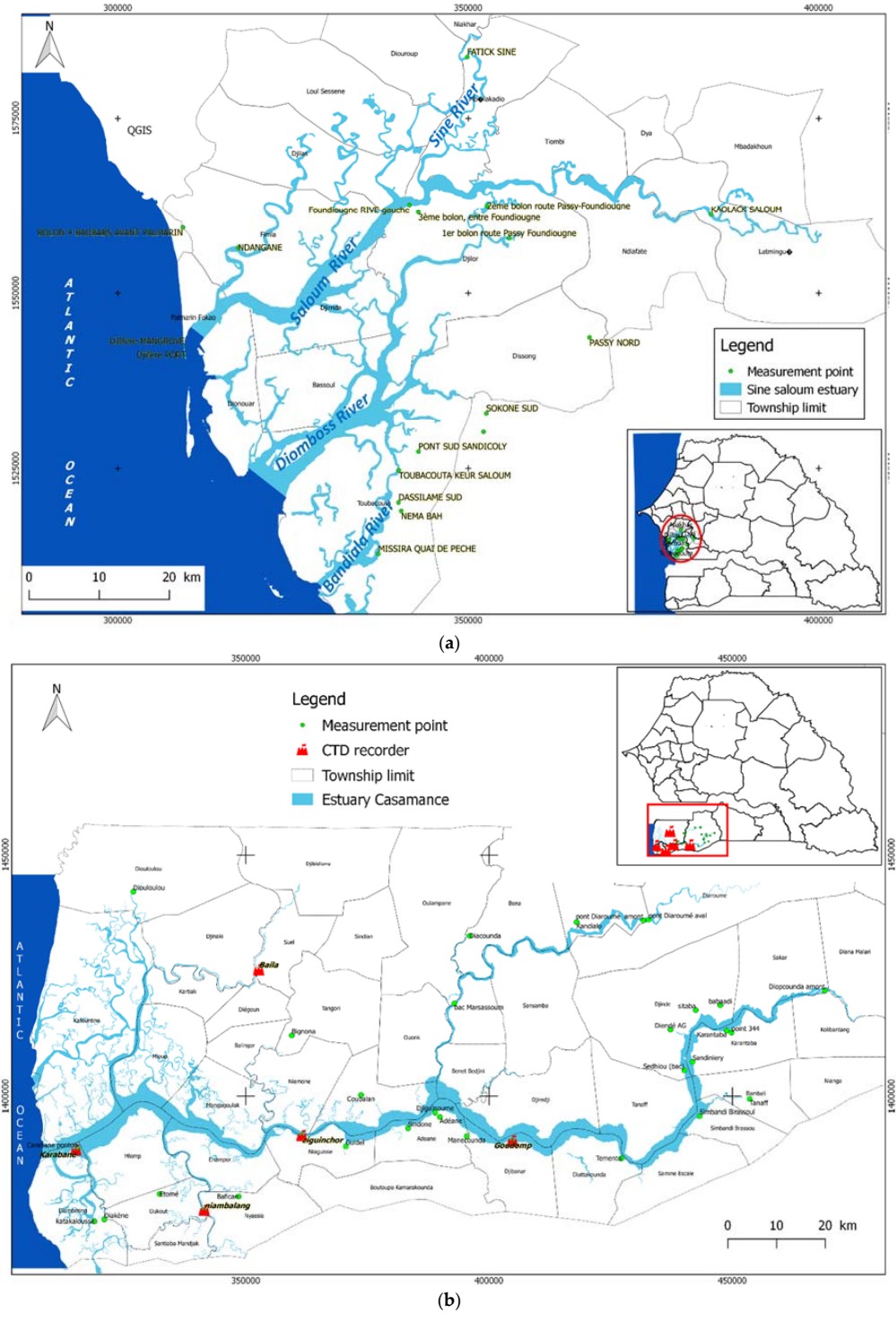

(**a**)

(**b**)

**Figure 4.** Location of devices and measurement points: (**a**) Saloum; (**b**) Casamance.

Despite the different periods of observation in each one of the settled devices, some general observations can be made about the current behavior of the salinity in the estuaries.

*3.1. Casamance Estuary*

Figure 5a–e show the seasonal evolution of water salinity in five points of the Casamance estuary from 2013 to 2019. The locations of the five salinity measurement stations are shown in Figure 4b.

- The salinity annual variation increases from the mouth (located 2 km downstream from Karabane station) to the upstream part of the estuary.

  ○ this variation increases in the main reach of the Casamance river, from Karabane (Figure 5a) to Ziguinchor (70 km from the mouth; Figure 5b) and then to Goudomp (125 km from the mouth, Figure 5c)

  ○ it also increases from the main branch to secondary branches, the "tributaries" coming from the North, at Baila on the "Baila bolon" (Figure 5d; *bolon* is the mandinka name given to the saline rivers of the mangroves in West Africa), and from the South, at Niambalang on the Kamobeul bolon (Figure 5e);

- The mean salinity values remain close to those of the sea (slightly above, at 40 g/L instead of 35 g/L) in the estuary (at least until Goudomp), as well as in the south branch of Kamobeul Bolon; it is significantly higher in the northern branch of the Baila Bolon (55 g/L).

Figure 6a–f shows the spatial variation of salinity at two stages of the year in the Casamance estuary. Figure 6a,c,e gives the salinity at the end of the rainy season in 2016, 2017, and 2018, respectively; Figure 6b,d,f gives the salinity at the end of the dry season, in 2017, 2018, and 2019, respectively.
The following characteristics are observed:

- There is along all the year a fresh water income at the upstream entry of the main branch of the Casamance estuary (at Diopcounda Bridge); the same observation is made at the upstream origin of its main tributary, the Soungrougrou (Diaroumé Bridge); however, fresh water discharge is significantly lower in this river;
- There is always an estuarine turbidity maximum (ETM, [36,37]) in the upper part of both the Soungrougrou and the Casamance;

  ○ This area moves upstream during the dry season and it reaches the highest salinity values of the main reach of the Casamance (70 g/L in the Casamance, 100 g/L in the Soungrougrou);

  ○ It moves downstream during the rainy season, pushed by the fresh water discharge coming from the (small) basin of the Casamance and Soungrougrou rivers. The salinity values decrease during this period;

  ○ Downstream of this moving peak, salinity decreases all year long; then, Casamance river has an inverse estuary, however, its upper part has a normal functioning during a few kilometers in the dry season and over some tens of kilometers in rainy season;

  ○ In the main branch (Casamance), a second salinity peak is observed during some seasons at the confluence with the Soungrougrou river, due to the upper salinity values of the latter;

  ○ As observed in Figure 4a–f, salinity values are lower in the rainy season and higher in the dry season in the tributary bolons than in the main reach of the Casamance river estuary.

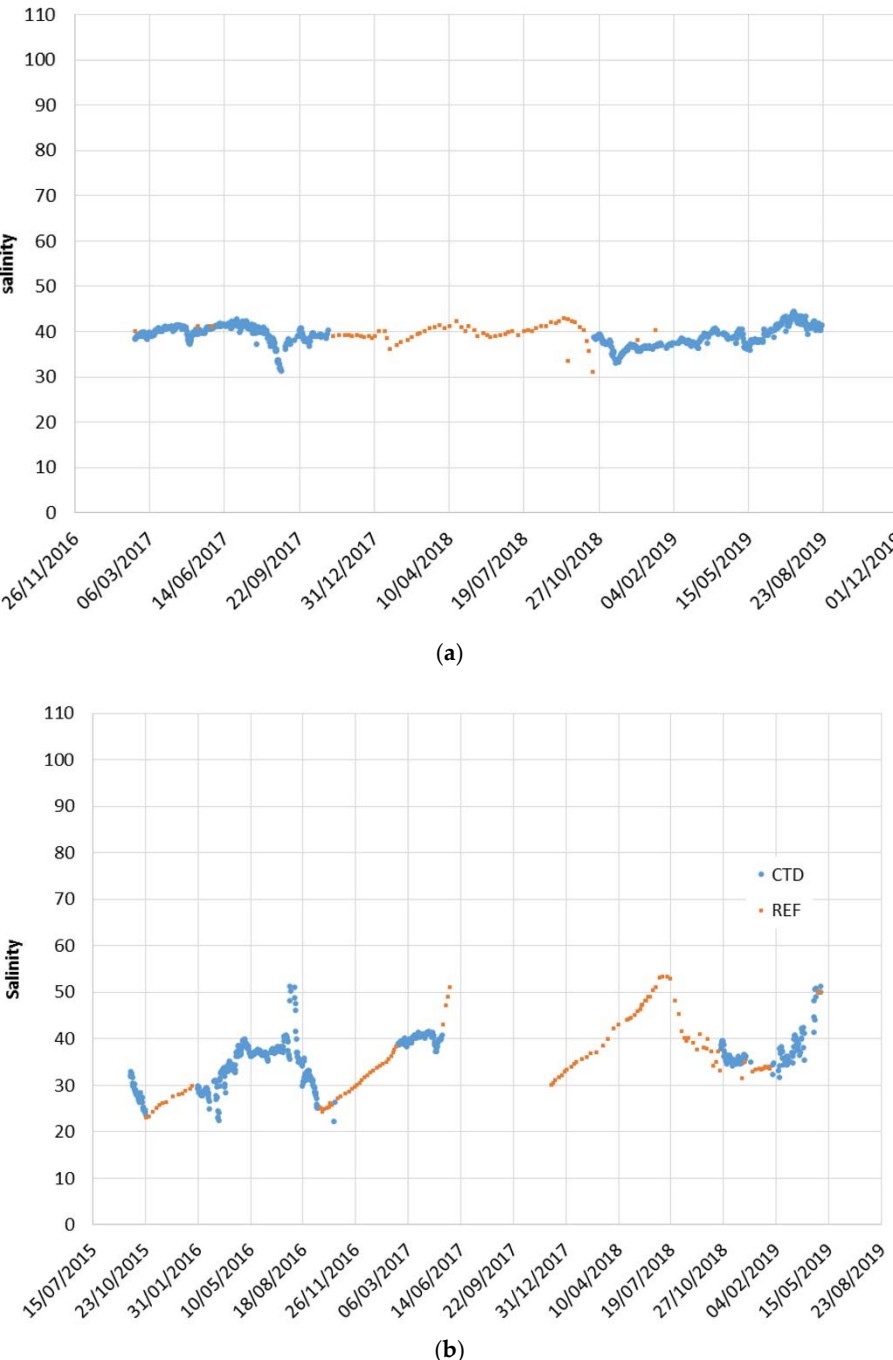

(**a**)

(**b**)

**Figure 5.** *Cont.*

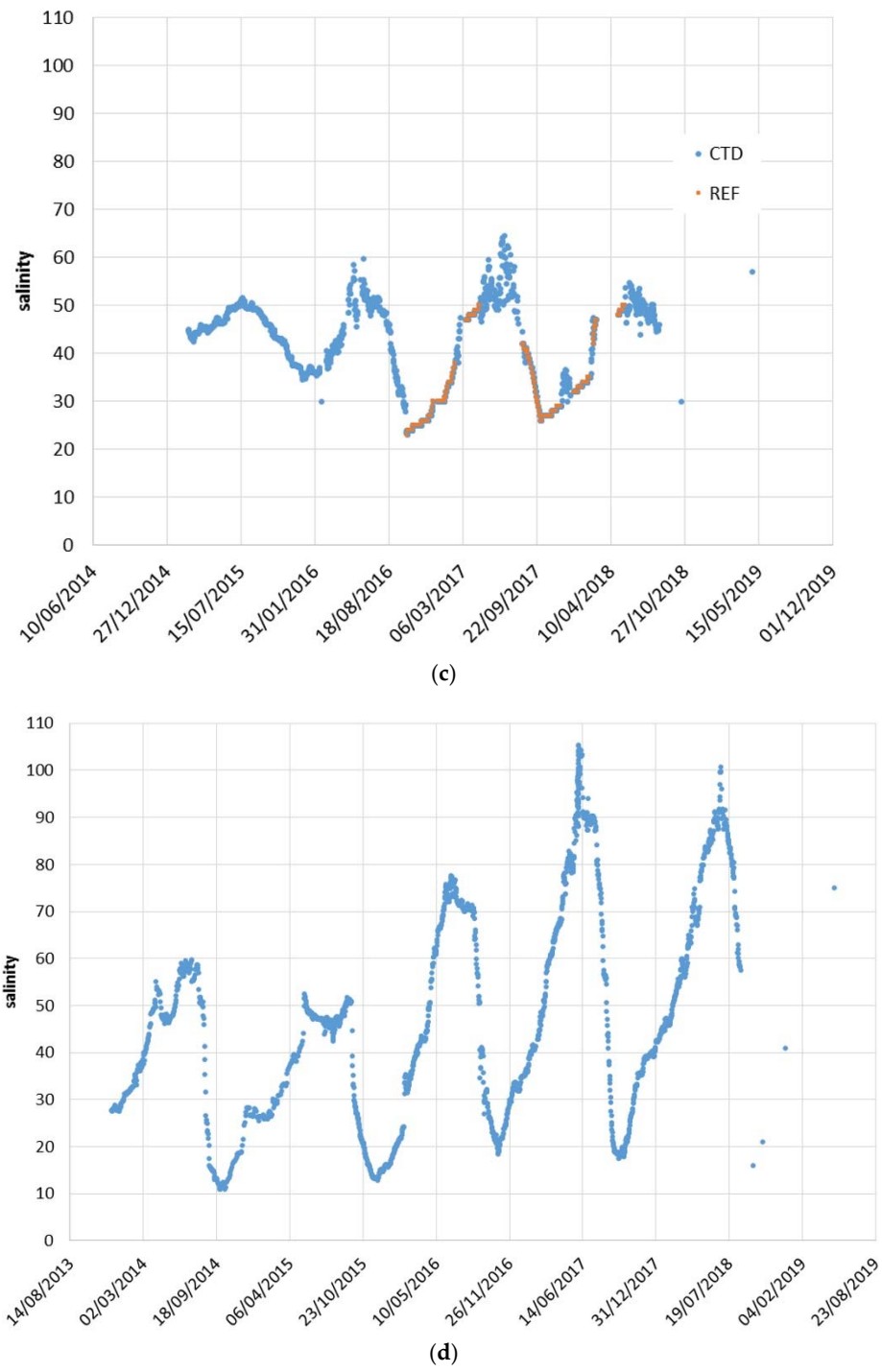

**Figure 5.** *Cont*.

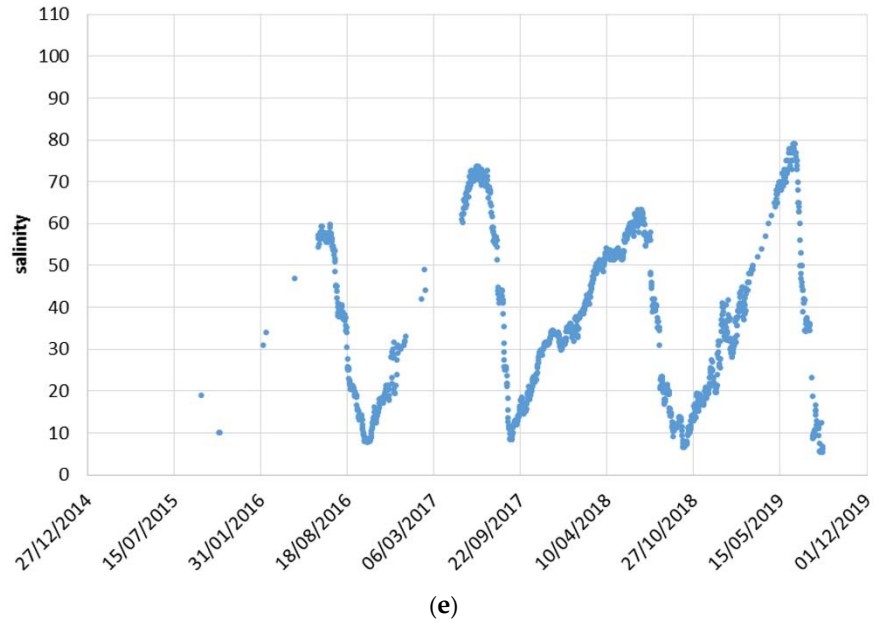

(**e**)

**Figure 5.** Evolution of salinity (g.L$^{-1}$) at the recording CTD stations: (**a**) Karabane (February 2017–August 2019); (**b**) Ziguinchor (September 2015–August 2019); (**c**) Goudoump (February 2015–October 2018); (**d**) Baila (December 2013–August 2018); (**e**) Niambalang (June 2016–August 2019). Red points are reconstructed data based on tens of water salinity measurement made by a refractometer.

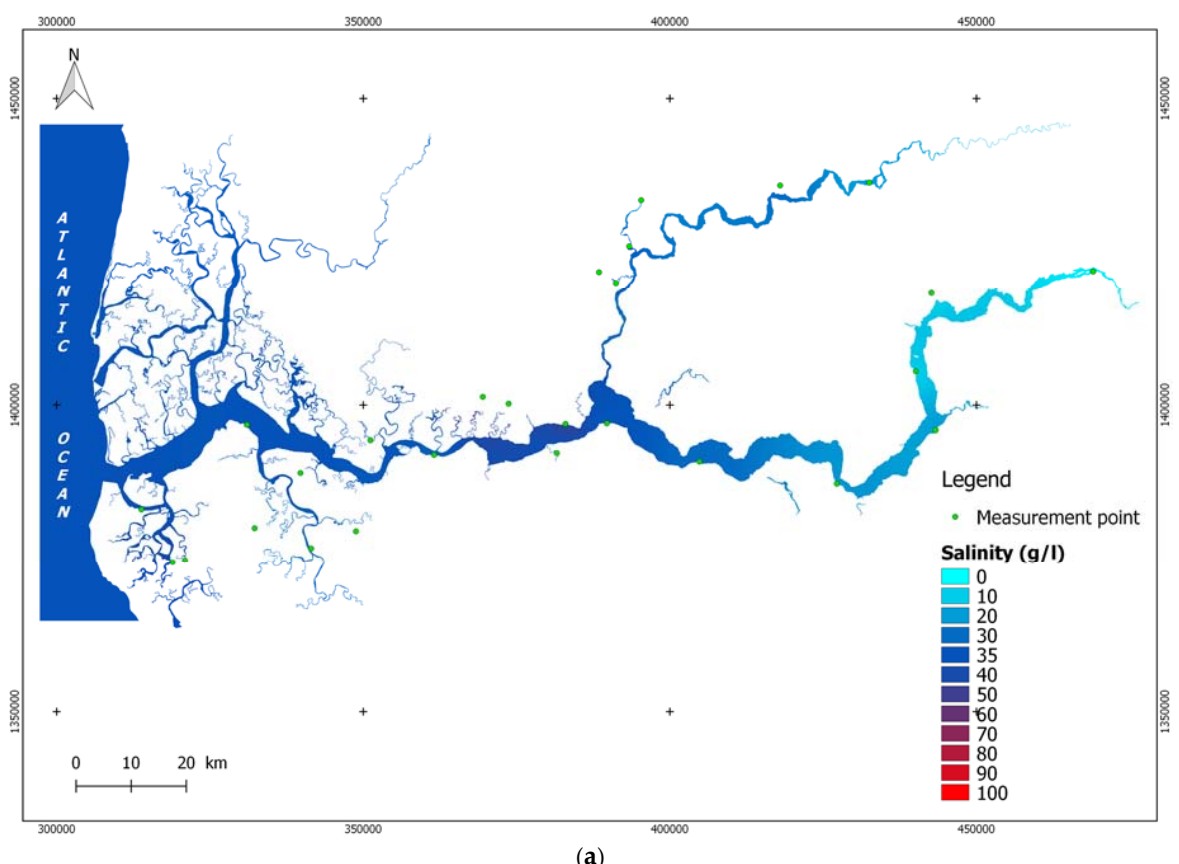

(**a**)

**Figure 6.** *Cont.*

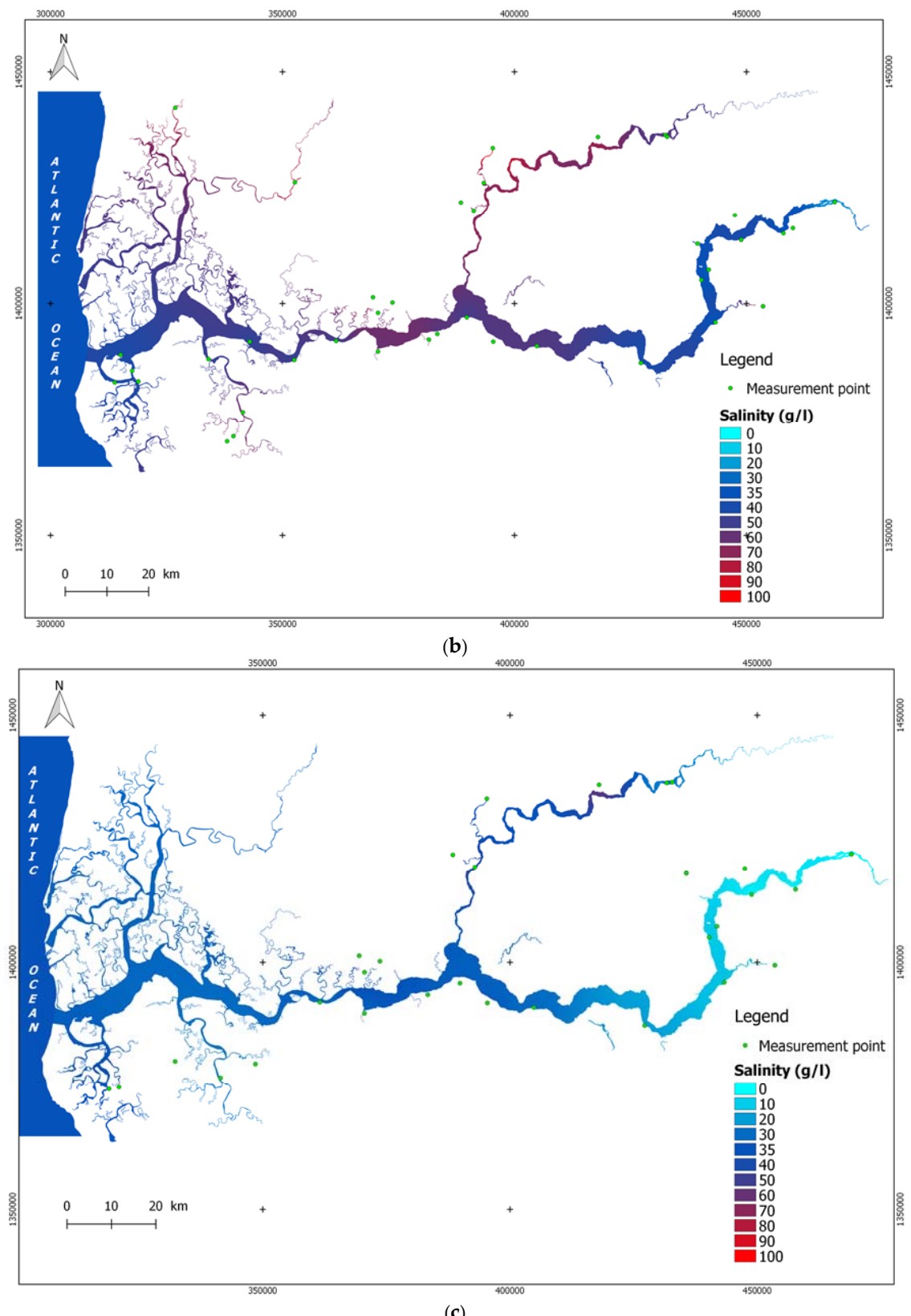

**Figure 6.** *Cont.*

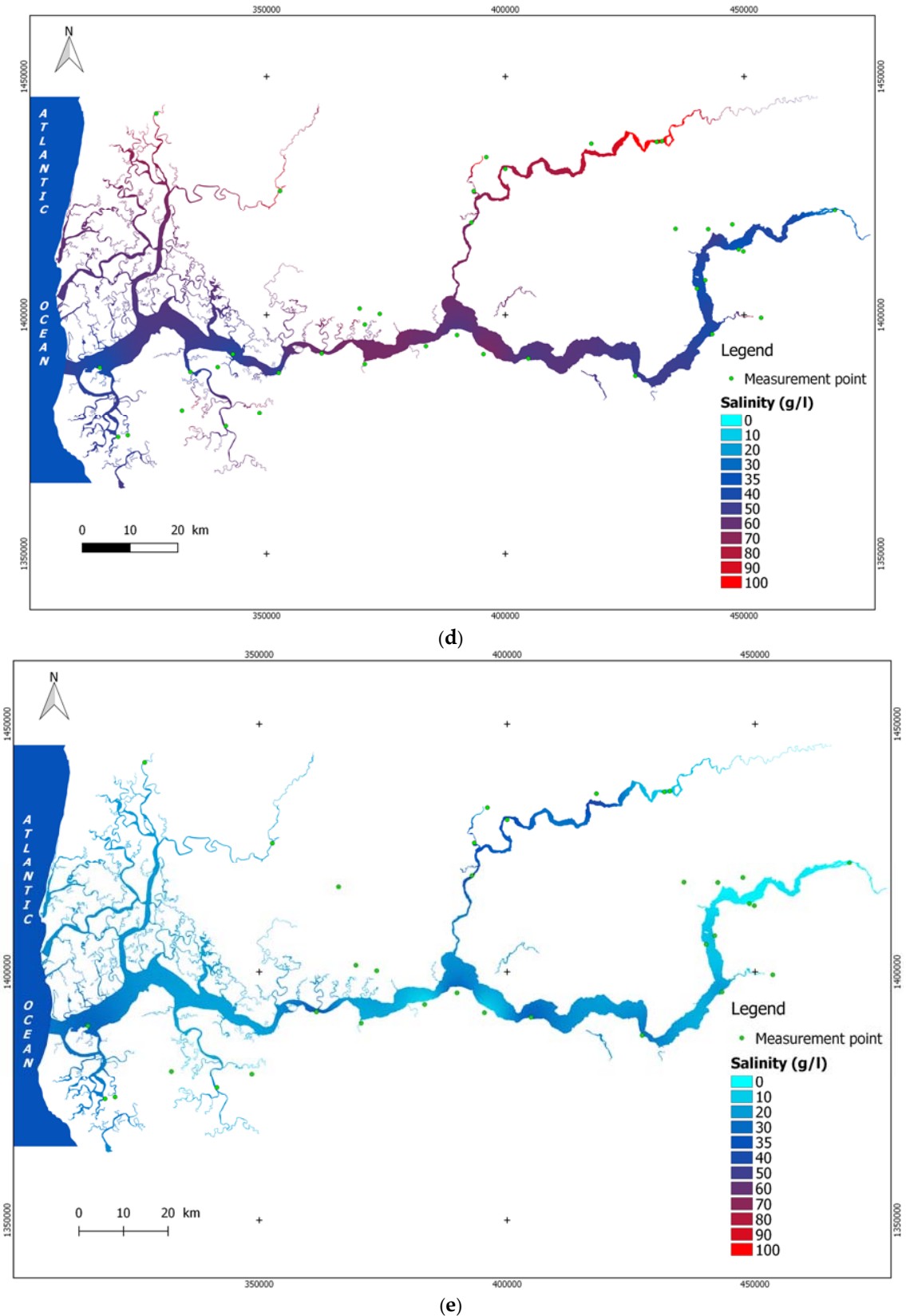

(**d**)

(**e**)

**Figure 6.** *Cont.*

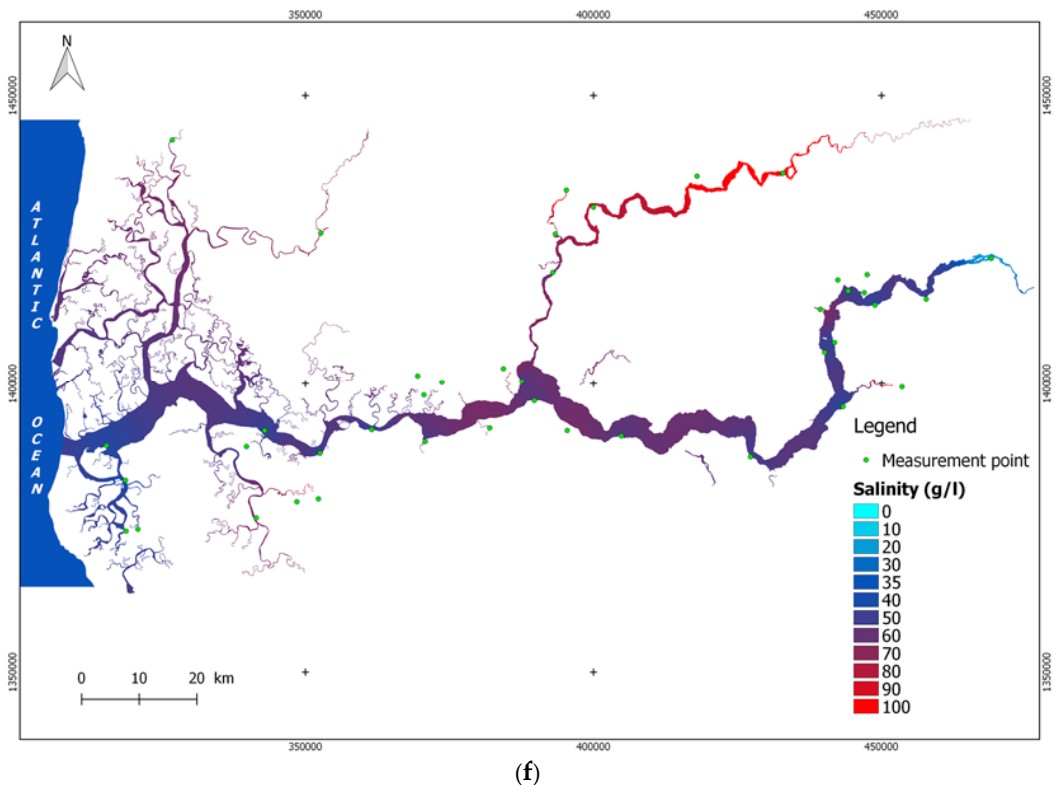

(**f**)

**Figure 6.** Casamance estuary salinity evolution: (**a**,**c**,**e**) at the end of the rainy season. (**a**) December 2016. (**c**) December 2017. (**e**). October 2018. (**b**,**d**,**f**), at the end of the dry season: (**b**) May 2017. (**d**) May 2018. (**f**) April 2019.

### 3.2. Saloum Estuary

Figure 7a–f documents the salinity of the Saloum estuary at the end of the rainy season (Figure 7a,c,e for 2016, 2017, and 2018, respectively) and at the end of the dry season (Figure 7b,d,f for 2017, 2018, and 2019 respectively). The functioning of this estuary is simpler than that of the Casamance.

- The fresh water discharge in the rainy season is quasi null and thus completely negligible;
- Values are lower during the rainy season due to lower evaporation, rain fallen within the wide estuary zone, and the sum of many small inputs by surficial runoff and small bolons;
- The estuary has an inverse behavior all year long;
- The salinity always increases upwards; the maximal values are always measured completely upstream, at Kaolack bridge in the Saloum and at Fatick Bridge in the Sine river;
- The salinity is higher in the north branch of Saloum estuary than that in the mid one (Diomboss) and overall than that in the southern one (Bandiala) (see location Figure 4a);
- The Bandiala bolon is provided in fresh water by the Nema Bah river, which is a small permanent fresh water river; water comes from the abundant water table of the southern Saloum plateau.

The temporal interannual variability is discussed in the last part of the discussion.

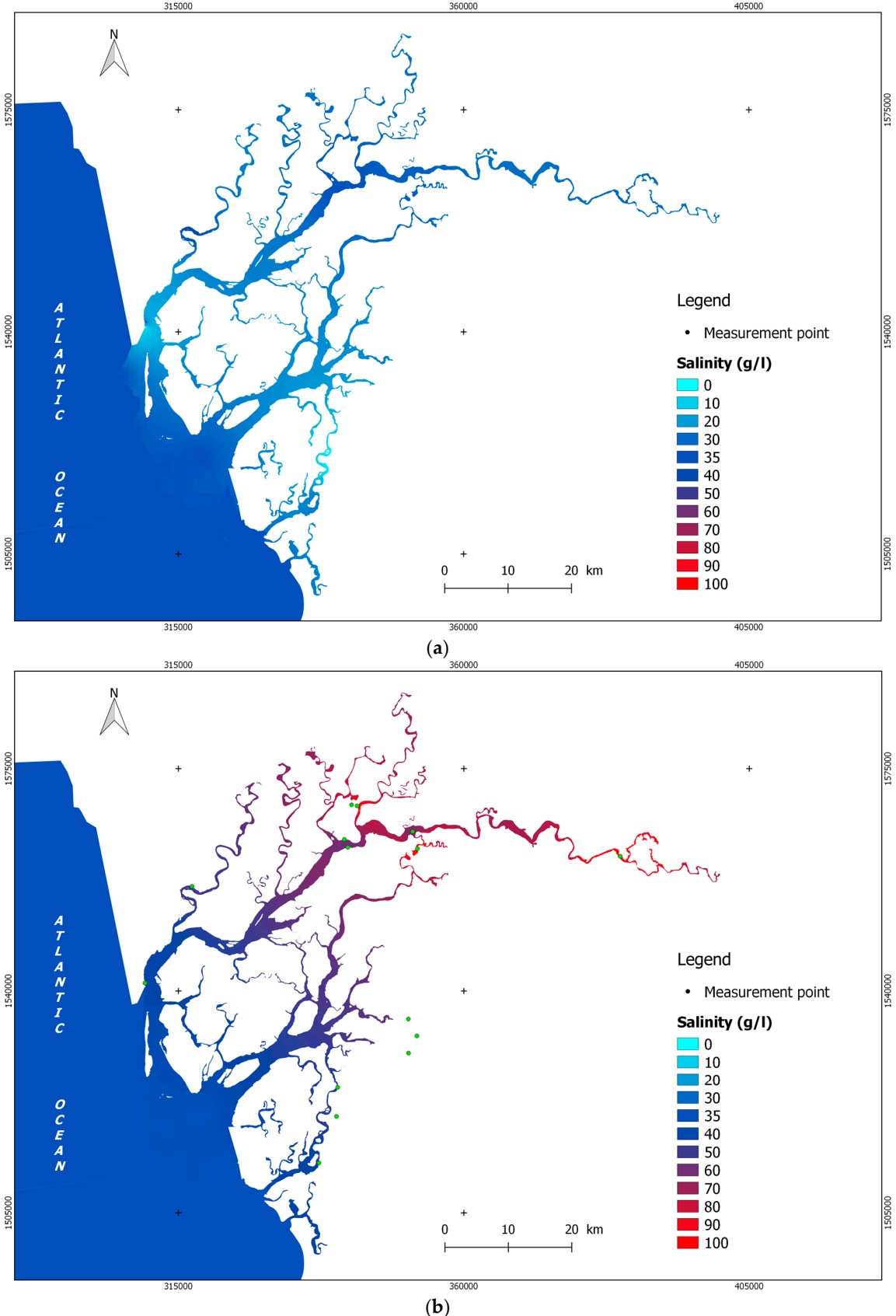

**Figure 7.** *Cont.*

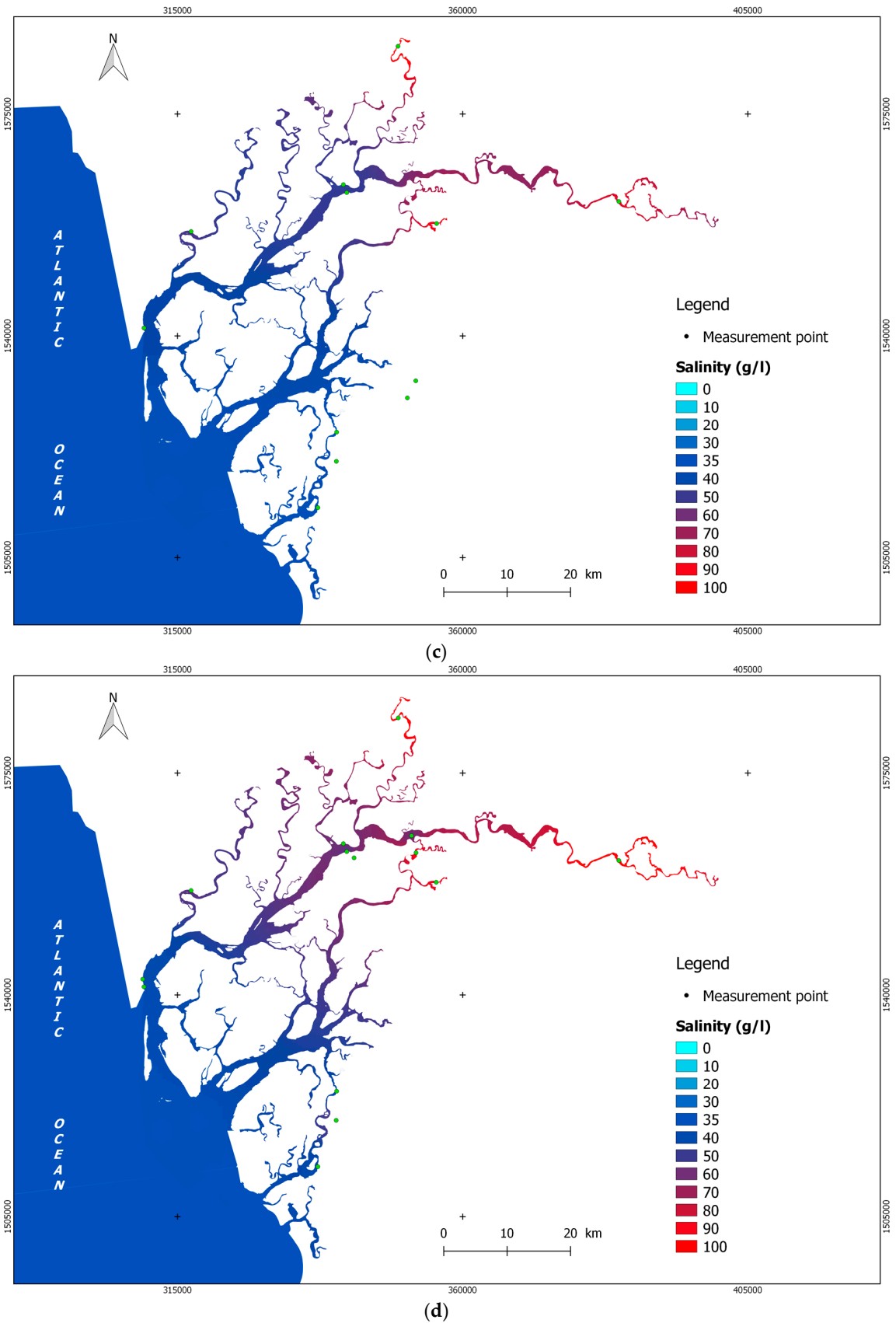

**Figure 7.** *Cont.*

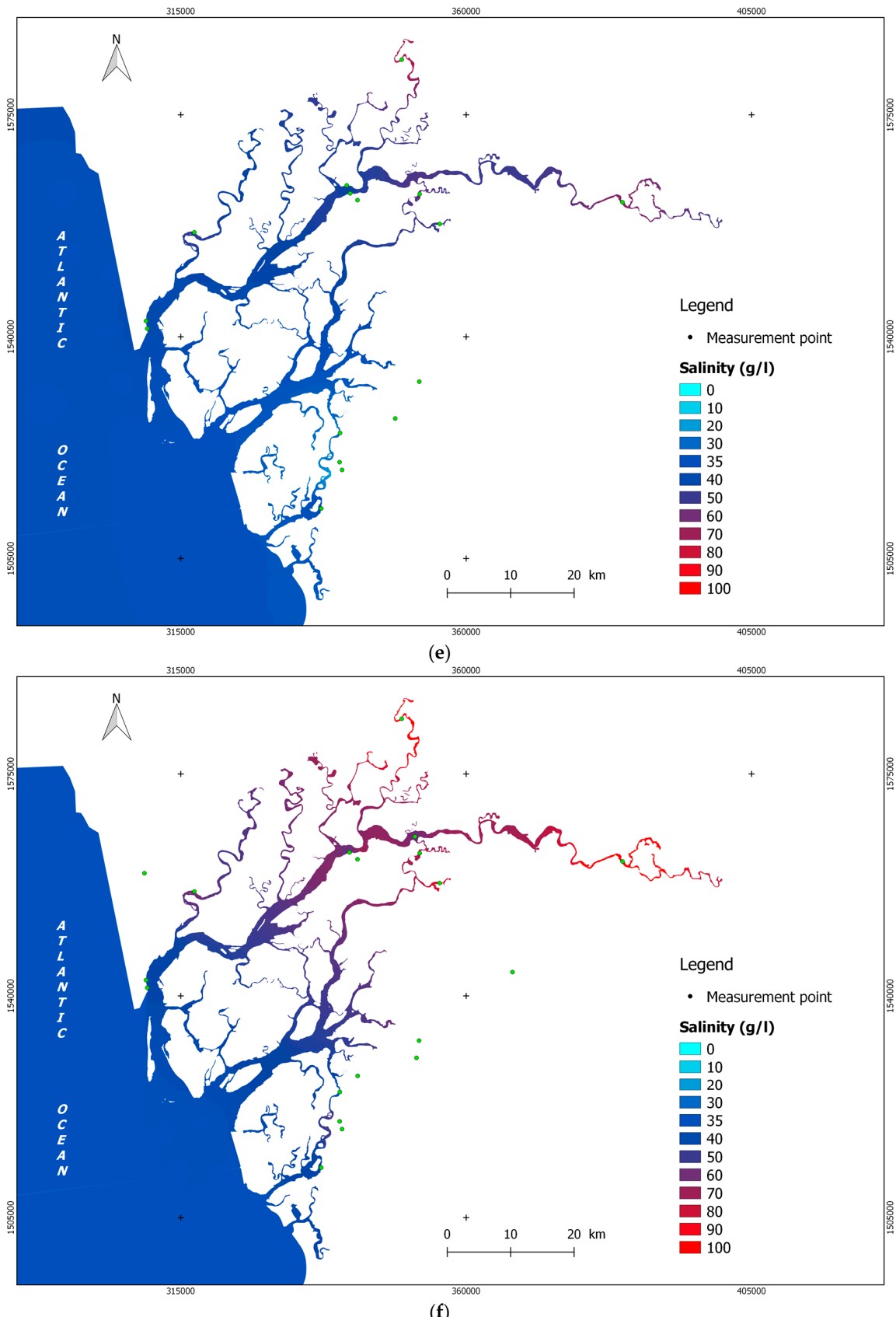

**Figure 7.** Saloum estuary salinity evolution: (**a**,**c**,**e**), at the end of the rainy season. (**a**) September 2016. (**c**) Jan 2018. (**e**) November 2018. (**b**,**d**,**f**), at the end of the dry season: (**b**) June 2017. (**d**) May 2018. (**f**) April 2019.

## 4. Discussion

### 4.1. A Comparison with Historical Data

One of the main references of past salinization (during the dry period 1968–1993) is the documentation about the Baila Bolon [3,38,39]. These authors highlighted the hypersalinization of the Baila bolon at its peak in 1984, as well as within the whole Senegambian valley [29]. In the Baila bolon, the mean salinity value was 96.8 g/L in 1983–1984; in the following years, salinity decreased until reaching 45.4 g/L in 1988–1989. Figure 8 shows the evolution of salinity at Baila (Baila bolon) during two periods. Salinization is not irreversible, and the rainfall recovery after 1988 allowed a quasi-complete desalinization at the end of the rainy season at the upstream sites of the estuary. The groundwaters have been recharged, and they flow through the salted rivers. They recharge the aquifers, rising water tables and allowing the conservation of all its fresh water [3].

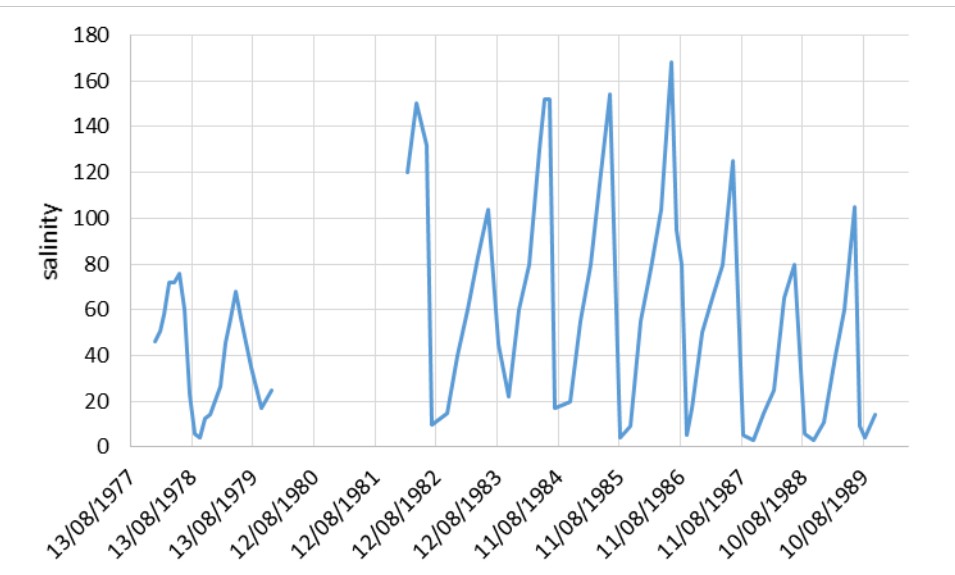

**Figure 8.** Salinity (g/L) measured at Baila during the Sahelian « great drought: 1978–1979 [32] and 1982–1989 [3].

The salinity at Baila station during the current period (Figure 5d), although presenting high interannual variability, is closer to the level observed during the 1978–1979 and 1988–1989 periods [3,32] than that measured during the 1982–1987 period [3]. The latter corresponds to the driest period of the Great Drought, the end of the second peak of dry years, which therefore is the period when the cumulated rainfall deficit was the highest. The 1978–1979 years are the period with a temporal rainfall recovery between the two "peaks" of droughts (1972–1974 and 1982–1984); the 1988–1989 period is also considered as the beginning of the rainfall recovery after the drought.

Table 2 shows the evolution of salinity at different points of the Casamance estuary, during two periods:

-       1978–1979 (in [32])
-       2016–2019 (our measurements)

**Table 2.** Salinity evolution at several places in Casamance estuary in 1977–1979 (Marius, 1985, p. 48, [32]) and in 2015–2019 periods (see location in Figure 4b).

| Date Salinity in g.L$^{-1}$<br>Place | January<br>1978 | May<br>1978 | November<br>1978 | May<br>1979 | November<br>2015 | May<br>2016 | November<br>2016 | May<br>2017 | November<br>2017 | May<br>2018 | November<br>2018 | May<br>2019 |
|---|---|---|---|---|---|---|---|---|---|---|---|---|
| Baila | 46 | 76 | 6 | 68 | 8 | 60 | 18 | 82 | 18 | 88 | 16 | 75 |
| Pointe St Georges | 44 | 48 | 30 | 56 | / | / | / | 41 | 35 | 40 | 31 | 43 |
| Etomé | 41 | 93 | 4 | 91 | 3 | 88 | 5 | 99 | 15 | 90 | 8 | 100 |
| Guidel | 53 | 89 | 0 | 88 | / | 50 | 16 | 65 | 32 | 65 | 17 | 55 |
| Marsassoum | 42 | 68 | 20 | 66 | / | / | 35 | 65 | 45 | 65 | 35 | 62 |
| Kandialo | 16 | 60 | 0 | 52 | / | / | 31 | 75 | 59 | 87 | 43 | 101 |
| Sedhiou | 8 | 30 | 5 | 33 | / | / | 10 | 35 | 9 | 38 | 10 | 45 |
| Diopcounda | 1 | 14 | 0 | 15 | / | / | 4 | 21 | 2 | 25 | 0 | 11 |
| Diaroumé | 30 | 60 | 13 | 63 | / | / | 20 | 56 | 27 | 120 | 10 | 126 |

The comparison of the two periods allow noticing some differences:

- The minimal salinity values are more pronounced during the first period; however, this is partially due to the fact that in the second period, only two measurements per year are made;
- Salinity increases between the first and the second period in all the upper valleys (Baila Bolon at Baila, Soungrougrou at Diaroumé and Kandialo, and Casamance at Diopcounda);
- It decreases only at the Guidel station;
- It remains approximately equal in the mid basins (Sedhiou in the Casamance, Marsassoum in the Soungrougrou) and the lower valley (Etomé and Pointe St Georges).

This suggests that in spite of the rainfall recovery, the salinity did not decrease completely after the second "peak" of the drought, during the 1980s.

Figure 9 allows the same conclusion about the Saloum estuary; values of salinity are higher in the 2016–2019 period than during the 1980–1982 one. It is likely due to the strong increase in salinity caused by the second peak of the drought (1983–1985); the rainfall recovery since the 1990s was not sufficient to complete a true desalinization.

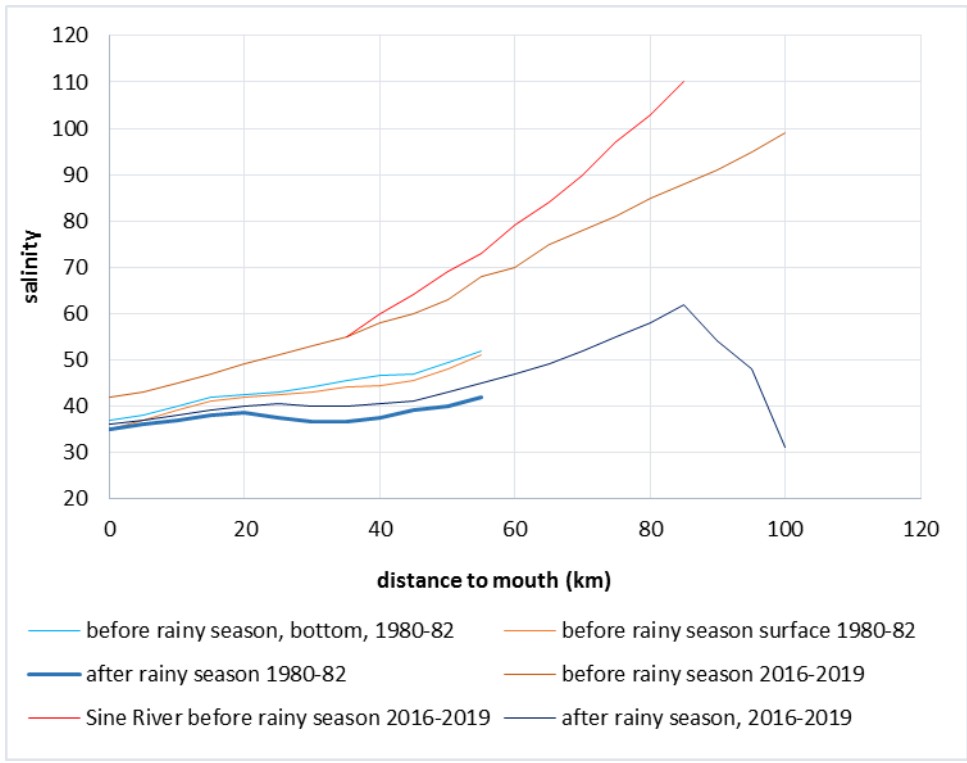

**Figure 9.** Salinity (g/L) vs. distance to mouth in the Saloum estuary, after [15].

Debenay and Pagès (1987) [40] defined the Casamance estuary, and they noticed five areas to be distinguished from downstream to upstream:

- From 0 to 50 km: a marine domain, with salinity, tides, and behavior close to those of the sea;
- From 50 to 85 km: an intermediary area, with increasing salinity;
- From 85 to 175 km: an hyperhaline area where salinity can reach 100 g/L;
- From 175 to 225 km: an alternative domain where salinity can vary from 0 to 100 g/L between rainy and dry season and during a few weeks only;
- Upstream from 225 km: fresh water with low discharge of the continental area.

The rainfall recovery in Casamance [25] and in the whole Senegambia [21] did not modify this general functioning; however, it caused a significant decrease in salinity in both estuaries of the Saloum and Casamance rivers.

### 4.2. An Integrative Indicator: The Mangrove

The mangrove forest is located at the boundary between the ocean and continent. The global mangrove area is declining mainly due to human activities [41,42]. Inversely, it is worth noticing that in West Africa, after a decline period during the Great Drought of 1968–1993, the mangrove area was significantly rising [43–45]; it almost reached its ancient extension level within the Saloum estuary and it even exceeds it within the Casamance estuary. It is expanding quickly [44,46] due to rainfall recovery and overall due to sea level rise. An improvement of mangrove governance and reforestation (unfortunately, mainly proceeded with Rhizophora in places where Avicennia was most indicated) locally could have contributed to this extension (only 4% of the increase in the mangrove area is due to reforestation, with the other 96% being spontaneous in the Saloum estuary [46]; respectively, these figures are 7% and 93% in the Casamance estuary [45]).

The mangrove is very sensitive to salinity levels; it can resist a few weeks or months to very high or very low salinity; however, if these peaks are repeated each year or if salinity values remain very high for more than 7 or 8 months a year, mangrove would eventually die.

Gilman et al. (2008) [47] describe the role of sedimentation on the mangrove stability and the ways that sedimentation may impact mangrove resilience (such as sediment accretion and erosion, biotic contributions, belowground primary production, autocompaction, fluctuations in water table levels, and pore water storage) but conclude that there is no correlation between sedimentation rates and sea level rise (SLR).

Osland et al. (2018) [48] show how the accelerated SLR could favor sedimentation and biotic contributions and thus the extension upland or upriver of the mangrove forest. Mangrove forests in arid and semiarid climates are known to be particularly vulnerable to changes in rainfall and freshwater availability [48]; the SLR probably accelerated the desalinization of the West African bolons, and this is probably the main explaining factor of the current mangrove expansion.

As an example, and contrary to the conclusion of Dièye et al. (2013) [49], the natural opening of the "fleche de Sangomar" sand spit in 1987, at the northern side of the Saloum estuary, allowed the entry of important volumes of marine water (35 g/L) in an hyperhaline estuary; then, it provoked a reduction of the salinity in the inland estuary. The mangrove regeneration was firstly observed near Djiffère in 1992 (Mamadou Sow, personal communication), where the opening of the "fleche de Sangomar" allowed a strong relative desalinization and the mangrove recovery. Mangrove decline in the lower Saloum estuary after 1987 and the end of the drought (rainfall annual total amount began increasing after 1985) observed by Dieye et al. (2013) [49] must be due to factors other than salinization.

SLR seems to be influencing the mangrove extension by leading to a relative desalinization of inverse estuaries water and by helping greater volumes of sea water entering in estuaries with much higher salt rates. The rainfall recovery since the end of the 1980s made the Casamance estuary show normal behavior during an increasing proportion of the year; this recovery did not allow at the time the Saloum River to have periods with normal functioning. However, there is no evidence that Saloum was not yet, before the drought, a completely inverse estuary at least since the African Humid Period, before 4000 BP.

The current expansion of the mangrove is the result of the significant relative desalinization of the inverse West African estuaries.

### 4.3. Low Discharges Explaining the Inverse Estuaries

Saloum and Casamance rivers have very low discharge values due to geology (sedimentary originated mainly sandy soils), the topography is very flat and, in addition, in Casamance, vegetation is very dense and rice cropping is organized to store most of the rainwater during the rainy season. Table 3 indicates the runoff coefficient values to be compared with that of surrounding basins.

Table 3. Mean rainfall, discharge, and runoff coefficient (KE) is the inverse estuaries, compared with their surrounding main basins.

| River Basin | Area km$^2$ | Annual Rain m$^3$ | Annual Discharge m$^3$ | % Total | Rain Depth mm | Mean Discharge m$^3$ s$^{-1}$ | KE % |
|---|---|---|---|---|---|---|---|
| Casamance at Diana Malari [1] | 4710 | 52,987 × 10$^6$ | 158,962 × 10$^3$ | 7.1 | 1125 | 5.08 | 3 |
| Soungrougrou [1] | 4480 | 51,609 × 10$^6$ | 51,610 × 10$^3$ | 2.3 | 1152 | 1.65 | 1 |
| Mid Casamance basin [1] | 4150 | 54,158 × 10$^6$ | 216,630 × 10$^3$ | 9.7 | 1305 | 6.93 | 4 |
| low Casamance basin Right bank [1] | 4323 | 56,631 × 10$^6$ | 283,156 × 10$^3$ | 12.7 | 1310 | 9.05 | 5 |
| low Casamance basin Left bank [1] | 1560 | 23,587 × 10$^6$ | 235,872 × 10$^3$ | 10.6 | 1512 | 7.54 | 10 |
| water body [2] | 927 | 12,857 × 10$^6$ | 1,285,749 × 10$^3$ | 57.6 | 1387 | 41.1 | 100 |
| CASAMANCE BASIN | 20,150 | 251,875 × 10$^6$ | 2,231,978 × 10$^3$ | 100 | 1250 | 71.4 | 8.9 |
| Nema Bah [3] | 50 | 318 × 10$^6$ | 2365 × 10$^3$ | 0.31 | 637 | 0.075 | 8.2 |
| Medina Djikoye [3] | 300 | 2145 × 10$^6$ | 10,373 × 10$^3$ | 1.38 | 716 | 0.33 | 6 |
| Car Car at Tataguine [4] | 1950 | 11,349 × 10$^6$ | 5674 × 10$^3$ | 0.75 | 582 | 0.18 | 0.5 |
| Sine at Fatick [4] | 3600 | 21,348 × 10$^6$ | 12,809 × 10$^3$ | 1.7 | 593 | 0.41 | 0.6 |
| Saloum at Kaolack [4] | 9502 | 61,180 × 10$^6$ | 91,770 × 10$^3$ | 12.19 | 644 | 2.93 | 1.5 |
| Lower basin [4] | 10,710 | 74,984 × 10$^6$ | 299,936 × 10$^3$ | 39.85 | 700 | 9.59 | 4 |
| Water Body [2] | 388 | 3298 × 10$^6$ | 329,800 × 10$^3$ | 43.82 | 850 | 10.55 | 100 |
| SALOUM BASIN | 26,500 | 174,625 × 10$^6$ | 752,727 × 10$^3$ | 100 | 659 | 24.07 | 4.3 |
| Senegal [5] | 337,000 | 2,527,500 × 10$^6$ | 20,183,040 × 10$^3$ | | 751 | 640 | 8.0 |
| Gambia [5] | 60,000 | 612,000 × 10$^6$ | 8,675,400 × 10$^3$ | | 1020 | 275 | 14.2 |
| Geba [6] | 12,440 | 164,208 × 10$^6$ | 1,970,496 × 10$^3$ | | 1320 | 62.5 | 12 |
| Corubal [2] | 26,000 | 445,640 × 10$^6$ | 10,249,200 × 10$^3$ | | 1714 | 325 | 23 |

[1] Dacosta, 1989 [50]; [2] calculated from GIS and rainfall data, + Guinea Bissau hydrological service for Geba River; [3] Mendy, 2010 [51]; [4] estimated from few existing data and double mass comparison with similar or surrounding basins; [5] [22]; [6] Sambou, 2019 [52]. See location of basins in Figure 3.

Clearly, runoff coefficients are lower in the Saloum and Casamance basins compared with other basins that have a comparable total rainfall amount. This is one of the main explaining factors of their inverse functioning.

The Casamance River basin has a runoff coefficient just slightly higher than that of the Senegal River basin, although it receives 55% more rainwater than the latter. The runoff coefficient of the Saloum River basin is half of that of the Casamance River. The Geba River basin also has a low runoff coefficient compared with that of the other basins with similar rainfall amount.

These three basins have little runoff due to geological and topographical factors. In the two first cases, runoff is significantly enhanced by the rain water fallen directly in the estuary; this constitutes 44% of the total discharge of freshwater in the Saloum basin and 58% of that for the Casamance basin.

## 5. Conclusions

These observations about salinity spatiotemporal evolution carried out within two West African inverse estuaries allow us to attempt a few statements:

Firstly, about the functioning of estuaries, we can confirm that:

- The Saloum River estuary has a total inverse behavior, with salinity increasing upwards;
- The Casamance estuary has a spatially partial inverse functioning, with a point of maximum salinity migrating from 20–30 km of the upstream end estuary at the end of the dry season to 50–80 km downstream from this point at the end of the rainy season.
- Therefore, about the spatial variability, we observed:
- decreasing salinity downwards from the peak of the ETM in the Casamance estuary and from the upstream entry of the estuary at Kaolack in the Saloum river;
- Increasing salinity seasonal variability in the tributary bolongs;
- A similar behavior in the bolongs than in the Casamance upper estuary.

Finally, the salinity seasonal variability increases with the distance to the ocean.

A relative desalinization is attested by the comparison of measurement realized since 2013. Besides, the progressive post-drought rainfall recovery led to a decreasing bolongs water salinity but answering with a 10–15-year delay. The mangrove spectacular recovery is a good indicator of the progressive reduction of the bolongs' water salinity.

However, the low discharge and runoff coefficient values in the continental part of the basins is one of the main and persistent explaining factors of this inverse functioning.

**Author Contributions:** Conceptualization, Y.S., M.T., S.-P.M., M.-J.S. and J.A.; Data curation, J.M., A.-B.D., D.S., Y.B., S.S. and A.D.(Awa Diop); Formal analysis, J.M., V.M., A.D. (Arame Dièye), A.B., A.-B.D. and B.F.; Funding acquisition, Y.S., M.T., S.-P.M., B.D.B. and J.M.; Investigation, Y.S., M.T., S.-P.M., B.D.B., J.M., S.C., A.D. (Arame Dièye), M.-J.S., D.S., S.S., A.D. (Awa Diop), B.F., B.A.S. and J.-P.V.; Methodology, V.M., S.C., A.D. (Arame Dièye), A.B., M.-J.S., A.-B.D., S.S., A.D. (Awa Diop) and B.F.; Project administration, B.A.S.; Resources, B.D.B., V.M. and S.C.; Software, A.B.; Supervision, E.M. and J.-P.M.; Validation, J.A. and J.-P.V.; Visualization, D.S. and Y.B.; Writing—original draft, L.D. Y.S. and M.T.; Writing—review and editing, L.D. S.-P.M., B.A.S., E.M., J.-P.M., J.A. and J.-P.V. All authors have read and agreed to the published version of the manuscript.

**Funding:** This research was funded by AFD (French Development Agency) (convention AFD CZZ 1894 01K).

**Acknowledgments:** The authors acknowledge the AFD (Agence Française de Développement) funded projet "Développement durable des zones littorales (Sénégal, Guinée Bissau, Guinée): vers une gouvernance citoyenne des territoires—(Sustainable development of coastal areas (Senegal, Guinea-Bissau, Guinea): towards a citizen governance of the territories (convention AFD CZZ 1894 01K), as well as the Grdr Migrations Citoyenneté Développement NGO for the project management.

**Conflicts of Interest:** The authors declare no conflict of interest.

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
