# Peer review of "Inverse Estuaries in West Africa: Evidence of the Rainfall Recovery?"

_water, doi:10.3390/w12030647_

Round 1
Reviewer 1 Report
I have revised the paper and I guess it is possible to accept after some changes. As general observation I have to state that figures have to been improved, some are not mentioned in the text at the right position, bibliographic references do not follow the journal’s rules, an abstract and keywords are missing and last, I miss some comparison between rain precipitation data and any teleconnection pattern (but this is just a suggestion), this could confirm the thesis proposed by Authors and make it more interesting to a wider public.
Specific comments:
Line 26, Rivers must be no capital
Line 51, erase comma
Fig 1, I guess it is not cited in the text and must be improved; variables on the “X” and “y” axis have to be specified in the graphics. Anyway this figure should go AFTER the figure 2, that is FIRST a location with river etc and than this, more specific figure,
The location has to INCLUDE a more wide map of ALL Africa to situate the reader
Line 104, “achivment” is not correct…maybe use “construction”
Line 107, is June and July…of which year? Or years?
Line 129-130, correct text size
Line 132-133 please specify the years during which the study was carried out
Line 144, you have number equations
Figure 3, you mix administrative and morphological criteria, not very clear
Figure 5 is not correctly reported, the figure caption has to be one and at the end of the figure, not repeated under each part
Figure 6, as above
Line 217, better say: along all the year
Line 268, erase “:”
Line 352, is it a sand spit? Please clarify
Line 378, River is in capital letter
Table 3 why is it there? Is an Annex? In that case it must cited in the right place and way
Author Response
Firstly, we warmly thank the reviewer to help us in improving our manuscript !!
The revised manuscript was revised by an native english speaker.
Reviewer 1:
Rev 1: Some figures are not mentioned in the text at the right position,
response to reviewer:
- Figures 1, 2, 3, 4, 5, 6 and 7 have been improved and the titles and numbers of figures 1, 2, 4, 5, 6, 7 corrected
- Figures 1 and 2 are now called in the text and figure 2 was changed in figure 1 to be the first one, explaining the position of estuaries; a map of Africa including Senegambia location was added
-
Rev 1: Bibliographic references do not follow the journal’s rules,
response to reviewer:
- References have been changed following the rules of the review
Rev 1: An abstract and keywords are missing.
response to reviewer:
- Abstract and keywords were included in the manuscript
Rev 1: I miss some comparison between rain precipitation data and any teleconnection pattern (but this is just a suggestion), this could confirm the thesis proposed by Authors and make it more interesting to a wider public.
response to reviewer:
- A reference (Mahé et al ,2013) was added to confirm teleconnection with the GCM and ITCZ patterns:
“ Teleconnections with the ITCZ (Inter Tropical Convergence Zone) fluctuations and their impact on flooding were analysed by Mahé et al. (2013) [26].”
Reviewer’s Specific comments:
response to reviewer:
- All the marking and comments were accepted and applied
- the figures number and references were changed (figures 5, 6 and 7)
- The Figure 3, the title of the maps was improved to better explain what is administrative and what is natural
Line 352, is it a sand spit? Please clarify
response to reviewer:
- yes the Pointe de Sangomar was a long sand spit, cut by a sea storm waves in 1987:
- “As an example, and contrary to the conclusion of Dièye et al. (2013) [49], the natural opening of the “fleche de Sangomar” sand spit in 1987, at the northern side of the Saloum estuary, allowed the entry of important volumes of marine water (35g/l) in an hyperhaline estuary; then it provoked a reduction”
Table 3 is not an annex; it has been better included in the discussion section
Reviewer 2 Report
This manuscript investigated salinity spatial and temporal evolution within two West African inverse estuarie based on the observed data. The results are interesting and useful for regional water management. I suggested revision before published in the journal and my comments are as follows:
Line143-148 Have you validated this equation based on the measured salinity? Please improve the resolutions of the figures in the manuscript.Author Response
Firstly, we warmly thank the reviewer to help us in improving our manuscript !!
The revised manuscript was revised by an native english speaker.
Reviewer 2
Rev 2: This manuscript investigated salinity spatial and temporal evolution within two West African inverse estuarie based on the observed data. The results are interesting and useful for regional water management. I suggested revision before published in the journal and my comments are as follows:
Rev 2: Line143-148 Have you validated this equation based on the measured salinity?
response to reviewer:
- The following sentence was added :
“Equation [1] was validated with the measured values with both refractometer and field conductimeter. Since the bolons’ water temperature always ranged between 21°C and 28°C, the deviation between measured and calculated values was low, rarely exceeding 2.5% (the highest observed difference was 4.8%).”
Rev 2: Please improve the resolutions of the figures in the manuscript.
response to reviewer:
- Resolution of figures 1 to 7 was improved